# Progressive relaxation training in patients with breast cancer receiving aromatase inhibitor therapy-randomized controlled trial

**Umut Bahçacı** [1]*, **Songül Atasavun Uysal**[2], **Zeynep Erdogan İyigün**[3], **Çetin Ordu**[4], **Gürsel Remzi Soybir**[5], **Vahit Ozmen**[6]

**1** Graduate School of Health Sciences, Physical Therapy and Rehabilitation, Hacettepe University, Ankara, Turkey, **2** Faculty of Physiotherapy and Rehabilitation, Hacettepe University, Ankara, Turkey, **3** Faculty of Medicine, Istanbul Bilim University, Istanbul, Turkey, **4** Department of Oncology, Gayrettepe Florence Nightingale Hospital, Istanbul, Turkey, **5** Department of General Surgery, Memorial Etiler Health Center, Istanbul, Turkey, **6** Breast Health Center, Istanbul Florence Nightingale Hospital, Istanbul, Turkey

* umutbahcaci@gmail.com

## Abstract

### Background

Aromatase inhibitors have positive impacts on the disease-free life of patients with breast cancer. However, their side effects, especially arthralgia, may be experienced by many patients. This study sought to assess the efficacy of Progressive Relaxation Exercises on the prevalent side effects of Aromatase Inhibitors in patients with breast cancer.

### Materials and methods

This clinical trial was conducted with single-blind randomization at a physiotherapy department in a local hospital. Patients who received Aromatase Inhibitor were assigned at random to either the study or control group. The study group (n = 22) performed a Progressive Relaxation Exercises program four days a week for six weeks, while the control group (n = 22) received advice on relaxation for daily life. Data was collected before the intervention and after six weeks. The study's primary endpoint was the Brief Pain Inventory, which was used to measure pain severity. Secondary endpoints included assessments of quality of life and emotional status, which were measured using the Functional Assessment of Chronic Illness Therapy and Hospital Anxiety and Depression scales, respectively.

### Results

The study group exhibited a significant reduction in Pain Severity (p = 0.001) and Pain Interference (p = 0.012) sub-scores. Reduction in Pain Severity (p<0.001) and Patient Pain Experience (p = 0.003) sub-scores was also noted between the groups. Quality of Life and Emotional Status showed no significant variation both within and between the groups (p>0.05).

**Data Availability Statement:** All relevant data are within the manuscript and its Supporting Information files.

**Funding:** The author(s) received no specific funding for this work.

**Competing interests:** The authors have declared that no competing interests exist.

## Conclusion

The study demonstrated that Progressive Relaxation Exercises caused a significant reduction in pain scores among Breast Cancer patients receiving Aromatase Inhibitors. While a decrease in pain during the 6-week period is valuable data, it is necessary to monitor the long-term effects of relaxation techniques.

## Introduction

Whilst Aromatase Inhibitor (AI) can have positive effects on a disease-free life, their usage may be limited due to side effects such as joint pain, stiffness, arthralgia and myalgia [1–3]. Almost 50% of patients receiving AI experience arthralgia, particularly as part of AI-induced musculoskeletal syndrome (AI-MSS) [4]. The most common musculoskeletal adverse events associated with AI-MSS are bone loss and arthralgia, which are primarily caused by estrogen deficiency [5]. The mechanism behind AI-induced arthralgia has been defined in several ways [6]. One is that reduced estrogen levels increase pro-inflammatory cytokines such as IL-6 and IL-1 in articular cartilage cells, causing joint pain and swelling [7]. The other mechanism is that the aromatase and estrogen receptors are expressed in the brain and the spinal cord analgesic system. This contributes to significantly reduced pain thresholds. This contributes to significantly reduced pain thresholds [8]. Additionally, MRI findings have revealed the pathophysiology of AI-MSS, which is associated with tenosynovial changes such as tenosynovial fluid accumulation and edema in the subcutaneous tissue that may cause pain in patients receiving AI [9, 10]. Moreover, patients receiving AI can experience side effects such as cognitive dysfunctions [11], anxiety and depression [12], sleep problems, and fatigue [13]. All of these adverse events can seriously affect patients' quality of life (QoL) and are the reason for discontinuation of therapy. Therefore, it is of paramount importance to modulate the quality of life of breast cancer patients by reducing the adverse events of AI [14, 15].

Besides pharmacological approaches, other interventions to reduce AI side effects include acupuncture, nutritional supplementation, relaxation techniques and physical exercise. Exercise interventions have been recommended as a means of alleviating the side effects of AI in patients with breast cancer (BC). These interventions include walking, aquatic exercise, strength training, bench press, leg press, seated row, and others [6, 16]. Aerobic, resistive or combined exercises are commonly used to improve the quality of life in BC patients undergoing AI treatment [17, 18]. These exercise interventions are mostly effective on reduction in pain and stiffness, improving in muscle strength, quality of life and pain threshold [16]. Only two studies, one involving yoga and the other Thai-chi, have examined the effectiveness of relaxation techniques on AI associated arthralgia [19]. The non-utilization of direct relaxation exercises to reduce AI side effects can be considered as a gap. However, there is inadequate evidence regarding the impact of relaxation techniques on treatment-related side effects in breast cancer patients.

Jacobson first described Progressive Muscle Relaxation Exercises (PRE) in 1938 [20]. PRE is a technique used in various studies to relax the whole body. It works by maximum contraction and relaxation of different muscle groups, sometimes accompanied by deep breathing [20, 21]. PRE is a widely used method with many current modifications for various health conditions [22, 23]. Physiological, perceptual, and behavioral positive findings of muscle relaxation were well-defined [21]. PRE is effective in breast cancer patients, providing improvements in upper limb function and reducing anxiety, as well as reducing the severity of chemotherapy

symptoms [24–26]. On the other hand, PRE appears to be effective in managing symptoms and increasing self-efficacy in lung and head-neck cancer patients [27, 28]. Considering the types of exercises, it can be inferred that patients may find it easier to perform PRE and therefore prefer it.

Exercise interventions such as aerobic and/or resistance exercises and implementation of Tai Chi and yoga seems to reduce adverse effects in patients receiving hormone therapy and improve QoL [16, 29]. However, the effects of PRE in patients with BC receiving AI is not well-known. This study aims to investigate the effects of PRE on arthralgia, quality of life, and emotional status in patients with BC receiving AI.

## Materials and methods

All procedures performed in studies involving human participants were in accordance with the ethical standards of the institutional and/or national research committee and with the 1964 Helsinki Declaration and its later amendments or comparable ethical standards. The Research Ethics Committee sanctioned the study, and written informed consent was obtained from all study participants. The study was approved by the Research Ethics Committee of the Bilim University (Protocol number: 23.07.2019/2019-15-02) and was registered on ClinicalTrials.gov with registration number 'NCT04163692' on 15.11.2019.

### Study design

A two-armed, assessor-blinded, randomized controlled study was conducted. BC survivors receiving AI were randomized to the study (PRE performed) or control group. Patients' medical records were sourced from the Breast Cancer Centre of the hospital and eligible patients were invited to participate by telephone and recruited meanwhile between 1-10-2019 and 1-9-2020. All assessments and exercise sessions took place within the Physiotherapy Department of the same hospital. Outcomes were evaluated at the beginning of the study and six weeks after the intervention. All participants received provided information on musculoskeletal pain and relaxation exercises.

The sample size was determined based on a previous study that mentioned the impact of yoga on functional outcomes in breast cancer survivors with aromatase inhibitor-associated arthralgia [30]. The mean and standard deviation data of the QoL parameter ☞mean (SD): 89,33(20,18) and 106,05(17,02) respectively] were used. The target sample size was determined with the aim of recruiting at least 20 participants per group, based on a power of 80% and a confidence interval of 95%, with an effect size of 0,88. Including estimated dropout rate of 30%, it is planned to recruit 26 patients in each group.

### Participants & randomization

Postmenopausal breast cancer patients who met the inclusion criteria were identified based on a search across the medical records of Breast Cancer Center. Inclusion criteria were as follows: having used AI for more than 6 months and less than 5 years, diagnosis of breast cancer stage 1–3, being a post-menopausal woman aged under 70 years, being hormone receptor-positive, and having received approval from their physician for participating in the PRE program. The participants were excluded from the study if they had difficulty in communication, neurological or orthopedic problems that could affect to make exercise, had any type of surgery during intervention, participated in any physical training in the previous six months or during the intervention, or had a diagnosis of lymphedema.

Among 120 patients with BC receiving AI, who were eligible as per the study criteria, 54 participants agreed to participate in the study voluntarily after phone call (Fig 1). Participants

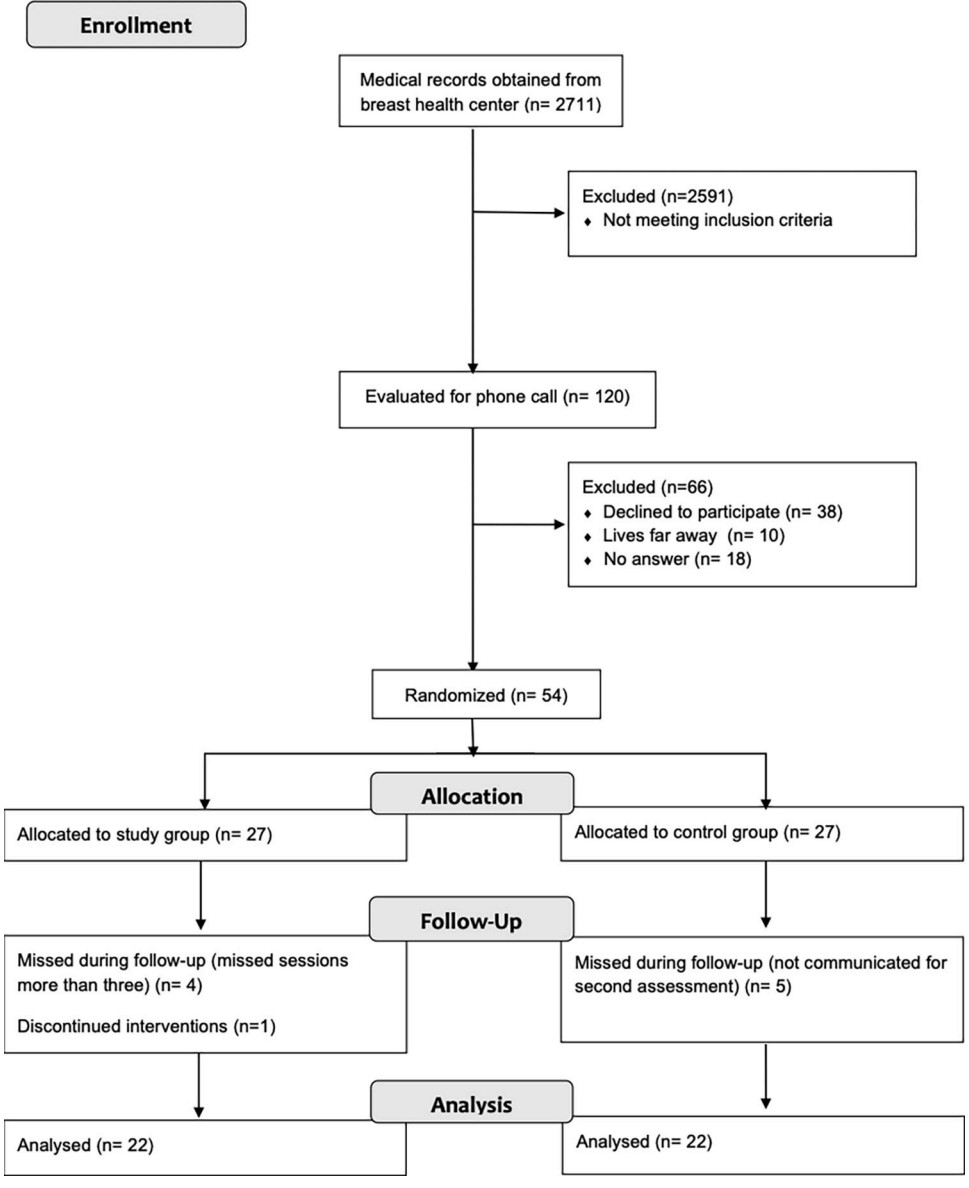

**Fig 1. Consort flow diagram.**

were invited for the baseline assessment at the hospital and the signed informed consent form was obtained from each patient before assessment.

Participants were allocated randomly to either the study group (n = 27) or the control group (n = 27) in a 1:1 ratio, utilizing a list of computer-generated random sequence numbers. The computer-generated list was used to check the last two digits of the participants' protocol number and determine the group. After this determination, the next participant was placed in the other group. Prior to randomization, an experienced researcher in oncologic rehabilitation, who was blinded to the allocation, evaluated patients for any musculoskeletal pain and other disorders. The patients were then asked to fill out the FACT, BPI and HAD forms. Following a 6-week intervention, the same assessor conducted a follow-up examination with the patients and had them complete the aforementioned forms once more (S1 Checklist and S1 and S2 Files).

## Outcome measures

**Pain.** As primary outcome, The *Brief Pain Inventory (BPI)* assesses the severity of pain and its impact on functioning. It has three dimensions of pain assessments as follows: pain severity (BPI-PS), pain interference (BPI-PI) and patient pain experience (BPI-PPE). The validated questionnaire consists of 17 questions, which evaluate the pain location, severity, and pain status, especially in the last 24 hours [31]. Higher scores are associated with higher pain [32, 33].

**Quality of life.** As secondary outcome, The *Functional Assessment of Chronic Illness Therapy—Breast (FACT-B)* instrument evaluates the all-round quality of life in patients with breast cancer and is available in many languages. The questionnaire includes five sub-scales that assess physical, social, emotional, functional and another anxiety status. It has a 27-item general (FACT-G) scale and a 10-item breast cancer-specific scale (FACT- B) in which patients evaluate their status. Patients determine an appropriate expression for themselves in the last seven days, with a 5-point scale as follows: 0: none; 1: very little; 2: slightly; 3: quite; 4: too much. Higher scores indicate a higher quality of life. The necessary permissions were obtained from the provider to access and use the validated version [34].

**Emotional status.** As secondary outcome, The *Hospital Anxiety and Depression (HAD)* scale evaluates the emotional status for physically ill patients. It consists of 14 questions 7 of which evaluate depression (HAD-D) and 7 of which consider anxiety (HAD-A). The cut-off score is 10/11 for the anxiety subscale and 7/8 for the depression subscale. Accordingly, any points above these scores are considered to be at risk [35].

**Intervention.** The PRE was implemented in the study group after the initial assessment in the hospital. The program was carried out four days a week, with one session being supervised and the remaining three sessions being carried out at home. The program was administered by the same physiotherapist and each session lasted for roughly 45–60 minutes. Participants were provided with a brochure detailing the exercise programs, which were identical between the supervised and home-based sessions. Participants were encouraged to continue the program independently. All queries related to home-based exercises were addressed during supervised sessions or via telephone calls. Each patient was given an exercise diary to maintain a regular routine as part of the program. Patients were advised to perform the exercises in a comfortable sitting position. Providing silence and a dim light helped to increase the effectiveness of the relaxation. Subsequently, the patients were presented with the instructions listed in Table 1 [21].

**Table 1. Instructions for PRE.**

| |
|---|
| • Clench your fists, push your elbow towards the seat |
| • Bend your elbows |
| • Push your shoulders back |
| • Press your knee down and pull your toes towards you |
| • Pull your knees towards you and push your feet down |
| • Tighten your hips |
| • Push your head back |
| • Lift your eyebrows |
| • Make wrinkles on your nose |
| • Tighten your teeth |
| • Push your chin down |
| • Close your eyes and think of good things. |

The patients completed each exercise with 5-second-long contractions and 10-second-long relaxation periods. They repeated each exercise three times. Breathing exercises were performed between the exercise periods to enhance relaxation effectiveness. Participants who missed over 10% of the sessions or failed to maintain the program were excluded.

The control group participants were advised to relax in their daily lives and avoid stressful living conditions. No supervised exercise or follow-up procedures were provided for the control group. Participants were advised that they could participate in PRE programs for six weeks if the study provides beneficial effects after intervention.

Both groups received information about the patients' progress and had their study-related inquiries addressed through phone calls, hospital sessions, or physician appointments.

## Statistical analysis

The SPSS software (Copyright ©SPSS Version 26) was used for statistical evaluation. The mean [95% confidence intervals (CI)], standard deviations, and frequency rates were summarized for baseline characteristics. The group distributions were examined using the Kolmogorov-Smirnov test. The group analysis was performed using the non-parametric test methods due to the low number of participants. The Mann-Whitney U test was used for the analysis of independent quantitative data, and Wilcoxon test was used for the analysis of dependent data. The McNemar test was used for dependent variables, and the Chi-Square test was used for independent variables for studying the qualitative data. Cohen's Formula was used for calculating the effect size of differences between and within the groups. The two-group t-test was used to calculate the sample size. The SPSS graphics were used to create figures; and a P value <0.05 was considered significant.

## Results

After 6-week intervention sessions, 44 out of 54 (%81.4) participants completed the assessments in both groups (Fig 1).

The groups were similar in terms of baseline characteristics, which were defined as age, body mass index (BMI), marital and educational status and type of AI used (p>0.05) (Table 2).

There was no difference in BPI, FACT and HAD scores in the initial values before the intervention for both groups (p>0.05).

There was a significant reduction in BPI-PS (p = 0.001) and BPI-PI (p = 0.012) scores between the initial and final values in the study group (Fig 2).

In contrast, the BPI-PPE, FACT-G, FACT-B, HAD-D and HAD-A scores between the initial and final values revealed no significance (all p>0.05) within the group's analysis (Table 3).

The analysis showed that the differences between the study and control groups were significant in terms of BPI-PS (p<0.001) and BPI-PPE (p = 0.003) scores. No significant differences were found between the groups in terms of BPI-PI, FACT-G, FACT-B, HAD-D and HAD-A scores (p>0.05) (Table 3).

## Discussion

The present study has been designed to shed light on the effect of PRE as an alternative therapeutic method to reduce pain and investigate its positive impact on the QoL and AD status in patients with BC receiving AI. Pain was significantly reduced within the study group and between groups in pain sub-scores. AI-MSS is a serious problem, which could be seen by up to 50% of patients with BC receiving AI [36]. Laroche et al. suggested that pain was a mostly psychological problem rather than being affected by genetic and biological factors [37]. Presant

**Table 2. Baseline characteristics of groups.**

| | | PRE group (n = 22) | Control group (n = 22) | p |
|---|---|---|---|---|
| Age (years), mean (SD) | | 60.5 (6.1) | 58.8 (8.1) | .58 |
| Body mass index (kg/m$^2$), mean (SD) | | 28.5 (4.09) | 27.7 (3.51) | .65 |
| Marital status (n-%) | Married | 16–72,7 | 19–86,4 | .26 |
| | Single | 6–27,3 | 3–13,6 | |
| Education status (n-%) | University | 14–63,6 | 9–40,9 | .12 |
| | High school | 3–13,6 | 9–40,9 | |
| | Primary education | 5–22,7 | 4–18,2 | |
| Type of AI (n-%) | Anastrozole | 12–54,5 | 14–63,6 | .54 |
| | Letrozole | 10–45,5 | 8–36,4 | |
| Breast Cancer Stage (n-%) | Stage I | 10–45,5 | 11–50 | |
| | Stage II | 8–36,4 | 8–36,4 | |
| | Stage III | 4–18,2 | 3–13,6 | .69 |

et al. reported that therapeutic management and patient's educational background were also significant factors for pain relief [36]. Our findings broadly support the work of other studies in this area linking to improve pain scores with PRE.

Pharmacological approaches have been identified as primary effective options to cope with AI-MSS in literature. At the same time, other complementary medicine strategies like acupuncture, nutritional supplementation, relaxing techniques and physical exercises are suggested as secondary options [19]. Some exercise methods were previously used to reduce pain and related emotional problems with limited success in AI users. Among these, aerobic exercise, resistance exercise or methods combining physical exercise with other methods seem to be efficacious techniques to cope with pain in the literature [17, 18, 38, 39]. The present study showed that, although the pain scores of the study group were higher than the control group before the intervention, it decreased even more after PRE. This shows us that they use the relaxation technique effectively to cope with pain. Improvements in the pain scores of the PRE group corroborate the literature findings.

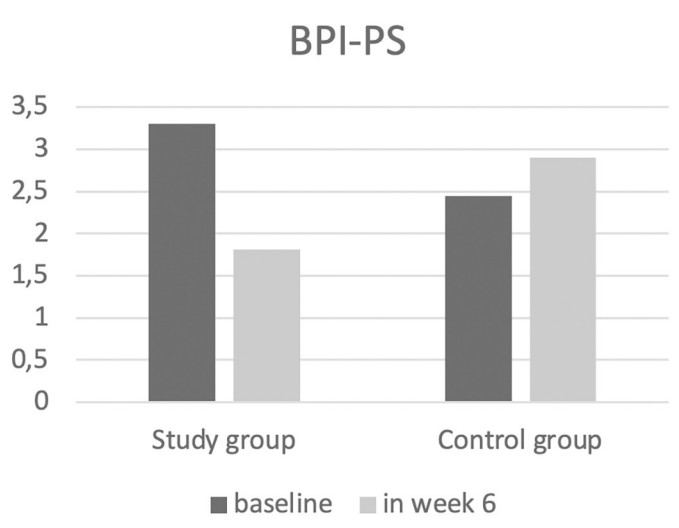
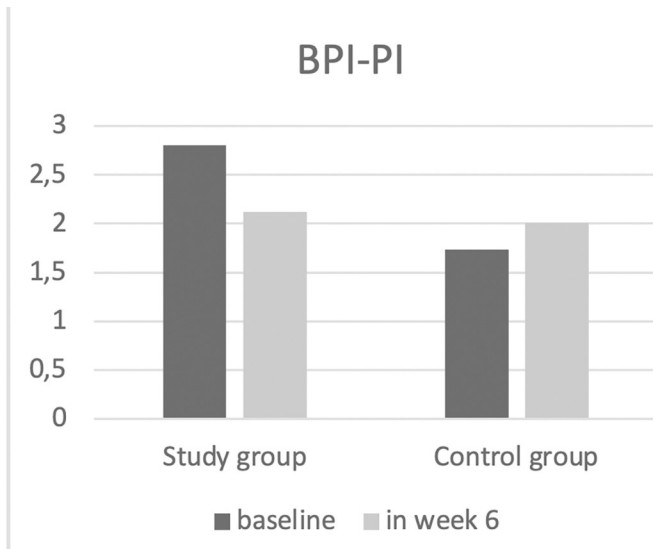

**Fig 2. Changes in BPI-PS and BPI-PI scores in groups.** BPI-PS: Brief Pain Inventory-Pain Severity, BPI-PI: Brief Pain Inventory-Pain interference.

**Table 3. Intra- and inter-group analysis for BPI, FACT, and HAD scores at baseline and Week 6.**

| Groups | | Study Group (n = 22) | | | Control Group (n = 22) | | | P* | R^ |
|---|---|---|---|---|---|---|---|---|---|
| Intervention outcomes | | Mean (SD*) | Median | 1Q-3Q | Mean (SD*) | Median | 1Q-3Q | | |
| BPI-PS | Baseline | 3.30 (1.81) | 3.00 | 1.94–4.38 | 2.44 (1.73) | 2.50 | 1.13–3.50 | ≤ .001[b] | 0.34 |
| | In Week 6 | 1.81 (1.49) | 1.50 | 0.69–3.38 | 2.90 (1.82) | 2.88 | 1.69–4.50 | | |
| | Effects within the groups (Baseline to Week 6) | 1.41 (1.58) | | | -0.46 (1.12) | | | | |
| | Significance within the groups (p) | **.001**[a] | | | .17 | | | | |
| BPI-PPE | Baseline | 3.05 (1.55) | 2.93 | 1.79–3.83 | 2.08 (1.78) | 2.21 | 0.78–2.96 | **.003**[b] | 0.99 |
| | In Week 6 | 2.02 (1.61) | 1.85 | 0.68–3.49 | 2.45 (1.73) | 1.84 | 1.03–3.48 | | |
| | Effects within the groups (Baseline to Week 6) | 1.03 (1.66) | | | -0.3 (1.12) | | | | |
| | Significance within the group (p) | .20 | | | .30 | | | | |
| BPI-PI | Baseline | 2.80 (1.86) | 2.61 | 1.21–2.62 | 1.73 (2.05) | 1.21 | 0.00–2.71 | .11 | 0.49 |
| | In Week 6 | 2.17 (2.07) | 1.85 | 0.12–4.14 | 2.01 (1.87) | 1.71 | 0.44–3.07 | | |
| | Effects within the groups (Baseline to Week 6) | 0.60 (2.21) | | | -0.27 (1.3) | | | | |
| | Significance within the group (p) | **.012**[a] | | | .16 | | | | |
| FACT-G | Baseline | 80.88 (11.83) | 79.50 | | 78.43 (19.90) | 84.00 | 61.83–94.66 | .89 | 0.10 |
| | In Week 6 | 82.92 (12.75) | 85.67 | | 81.34 (16.71) | 85.00 | 70.45–93.51 | | |
| | Effects within the groups (Baseline to Week 6) | -2.04 (9.41) | | | -2.91 (7.5) | | | | |
| | Significance within the group (p) | .30 | | | .27 | | | | |
| FACT-B | Baseline | 106.36 (17.56) | 107.00 | | 104.40 (26.07) | 109.93 | 84.83–125.07 | .88 | 0.09 |
| | In Week 6 | 108.86 (16.52) | 111.67 | | 107.88 (21.92) | 112.50 | 82.83–121.37 | | |
| | Effects within the groups (Baseline to Week 6) | -2.5 (12.2) | | | -3.5 (9.17) | | | | |
| | Significance within the group (p) | .43 | | | .24 | | | | |
| HAD-A | Baseline | 6.59 (3.1) | 7.00 | 4.00–9.00 | 6.91 (4.8) | 6.50 | 2.75–10.25 | .42 | 0.17 |
| | In Week 6 | 6.00 (3.00) | 6.00 | 4.00–8.25 | 6.73 (4.40) | 5.50 | 2.75–10.25 | | |
| | Effects within the groups (Baseline to Week 6) | 0.50 (2.3) | | | 0.18 (2.3) | | | | |
| | Significance within the group (p) | .56 | | | .82 | | | | |
| HAD-D | Baseline | 3.41 (2.3) | 3.00 | 1.00–5.00 | 4.50 (3.9) | 2.50 | 1.00–8.25 | .42 | 0.03 |
| | In Week 6 | 3.05 (2.88) | 2.50 | 0.75–5.00 | 4.23 (3.6) | 3.50 | 1.00–7.00 | | |
| | Effects within the groups (Baseline to Week 6) | 0.36 (2.9) | | | 0.27 (2.7) | | | | |
| | Significance within the group (p) | .28 | | | .83 | | | | |

BPI-PS: Brief Pain Inventory-Pain Severity, BPI-PPE: Brief Pain Inventory-Patient Pain Experience, BPI-PI: Brief Pain Inventory-Pain Interference, FACT-G: Functional Assessment of Chronic Illness Therapy-General, FACT-B: Functional Assessment of Chronic Illness Therapy-Breast, HAD-A: Hospital Anxiety Depression-Anxiety, HAD-D: Hospital Anxiety Depression-Depression, 1: standard deviation, a: Wilcoxon Signed-Rank test p<0.05, b: Mann-Whitney U Test p<0.05

*: Significance between the groups, ^: Effect size between the groups, data are reported as means of (%95 CI) and (SD)

Yoga and Tai-Chi are reported to be relaxation techniques, which are recommended to improve wellbeing and reduce pain in patients with BC receiving AI [30, 40]. PRE aims to relax most of the muscles in the form of a 'contract and relax' method, which is distinct from Tai-Chi and Yoga. PRE was performed as an exercise method rather than a technique to achieve muscle relaxation in the present study. Integrative therapies are recommended to reduce adverse effects of treatments, although more substantial evidence is needed to achieve a

consensus for wider use [41, 42]. PRE should be considered as an integrative therapeutic method to reduce pain in patients with BC receiving AI.

The present study did not find a significant difference in QoL and anxiety-depression (AD) outcomes with the implementation of PRE. There is no consensus in the literature on the positive effect of exercises for QOL and AD in patients with BC. Some of the literature studies suggest exercise to improve QoL and AD in patients with BC receiving AI [43–45]. In particular, the aerobic and resistance training seem to be effective in improving QoL and AD in AI-receiving patients with BC [38, 39]. On the other hand, Cadmus et al. showed that exercise did not affect QoL in either recently diagnosed or post-treatment in patients with BC [46]. Therefore, QoL and ES seem to need more substantial evidence among treatments of patients with BC.

The randomized design and a high rate of technical compliance of the patients with the scheduled exercise program constitute the power of this study. Furthermore, the high rate of (81%) adherence of the patients to the program was satisfactory during the present study. In contrast, the peer support in the intervention group, which can affect intervention results positively or negatively, can be considered a limitation. The other limitation is that the baseline BPI scores were higher in the study group than in the control group. This is noteworthy because it is important that even a 1-point difference can affect significance. However, the effect size between groups is low in both 'BPI-PS' and 'BPI-PI' scores. Long-term follow-up procedures with an increased sample size could be meaningful for future studies to ensure patient compliance with complementary therapies. Although telephone reminders and exercise diary keeping can improve fidelity to home exercises in this study, the use of fully supervised relaxation exercises will give more meaningful results. Further studies are needed to reach a consensus to recommend PRE as one of the primary and long-term treatment options for reducing AI-related pain in patients with BC.

In conclusion, this study indicate that supervised and home-based PRE could be efficient in reducing the pain sub-scores (PS and PPE) of patients with BC receiving AI. PRE intervention is not enough to affect the QoL and AD scores.

## Supporting information

**S1 Checklist. CONSORT 2010 checklist of information to include when reporting a randomised trial*.**
(DOC)

**S1 File. Study protocol original.**
(DOCX)

**S2 File. Study protocol English.**
(DOCX)

## Acknowledgments

Thanks to the medical oncology department at Florence Nightingale Hospital for their support and the individuals who participated in the study.

## Author Contributions

**Conceptualization:** Umut Bahçacı, Songül Atasavun Uysal, Zeynep Erdogan İyigün.

**Formal analysis:** Zeynep Erdogan İyigün.

**Investigation:** Umut Bahçacı, Zeynep Erdogan İyigün.

**Methodology:** Songül Atasavun Uysal.

**Project administration:** Songül Atasavun Uysal, Çetin Ordu, Gürsel Remzi Soybir, Vahit Ozmen.

**Resources:** Umut Bahçacı, Vahit Ozmen.

**Supervision:** Songül Atasavun Uysal.

**Visualization:** Songül Atasavun Uysal.

**Writing – original draft:** Umut Bahçacı.

**Writing – review & editing:** Umut Bahçacı, Songül Atasavun Uysal, Zeynep Erdogan İyigün, Çetin Ordu, Gürsel Remzi Soybir, Vahit Ozmen.

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
