## [Decision Letter · Decision Letter 0]

19 Dec 2023

PONE-D-23-36160Progressive Relaxation Training in Patients with Breast Cancer Receiving Aromatase Inhibitor Therapy-Randomized Controlled TrialPLOS ONE

Dear Dr. BAHÇACI,

Thank you for submitting your manuscript to PLOS ONE. After careful consideration, we feel that it has merit but does not fully meet PLOS ONE’s publication criteria as it currently stands. Therefore, we invite you to submit a revised version of the manuscript that addresses the points raised during the review process.

We look forward to receiving your revised manuscript.

Kind regards,

Peter Vuylsteke, MD

Academic Editor

PLOS ONE

Additional Editor Comments:

This is an interesting article and hypothesis, and the results are food for thought. Congratulations for the work done.

Still the comments made by the reviewers are valid and need to be adressed and responded to in a satisfactory way, before going further in the process.

Reviewers' comments:

Reviewer's Responses to Questions

**Comments to the Author**

1. Is the manuscript technically sound, and do the data support the conclusions?

Reviewer #1: Yes

Reviewer #2: Yes

Reviewer #3: No

2. Has the statistical analysis been performed appropriately and rigorously? 

Reviewer #1: Yes

Reviewer #2: Yes

Reviewer #3: No

3. Have the authors made all data underlying the findings in their manuscript fully available?

Reviewer #1: Yes

Reviewer #2: Yes

Reviewer #3: Yes

4. Is the manuscript presented in an intelligible fashion and written in standard English?

Reviewer #1: Yes

Reviewer #2: Yes

Reviewer #3: No

5. Review Comments to the Author

Reviewer #1: A two-arm randomized controlled clinical trial was conducted which aimed to assess the efficacy of Progressive Relaxation Exercises on the side effects of aromatase inhibitors in patients with breast cancer. The intervention group reported significantly less pain severity and pain interference than the controls. No differences in quality of life and emotional status were found.

Minor revisions:

1- P-values never equal zero; express small p-values as < 0.001.

2- Line 122: Provide full details of the sample size calculation. The power calculation should include: (1) the estimated outcomes in each group; (2) the α (type I) error level; (3) the statistical power (or the β (type II) error level); (4) the target sample size (5) statistical testing method and (6) for continuous outcomes, the standard deviation of the measurements.

3- Line 140: Indicate if block randomization was used. If so, specify the block size.

4- Line 212: For improved clarity, replace “analyzed” with “summarized.”

5- Table 3: When data is not normally distributed, it is customary to summarize the results using median, first and third quartiles.

Reviewer #2: This -to my knowledge-is the first study testing 'Progressive Relaxation Exercises' in patients receiving an oral aromatase inhibitor in early breast cancer. The study is positive for some endpoints (Pain); the authors do caution their conclusions saying that longer follow-up is needed and that PRE has no effect on other studied endpoint.

Small comments

- Please use the offical term 'AIMSS' troughout the manuscript as this is the name given to this well known problem (aromatase inhibitor musculoskeletal syndrome)

- Please describe the pathophysiology of AI-MSS (refering to fluid retention in the joints as studies by MR imaging and ultrasound); Morales et al. J Clin Oncol 2008

- The authors report on mean and SD; What clinicians do want to know is whether women (in one group) with more pain at baseline were more likely to respond to PRE/no-PRE than those with less pain at baseline.

- Some minor edits to be corrected in the text

Reviewer #3: The authors, Bahcaci et al. present data from their RCT on “Progressive Relaxation Training in Patients with Breast Cancer Receiving Aromatase Inhibitor Therapy-Randomized Controlled Trial,” which addresses an important area of morbidity for a large proportion of breast cancer patients treated with AIs, however the small sample size and the lack of analytical rigor limit clinical interpretation of this data.

The specific comments are list below. Applicable comments for the abstract are expanded upon below:

Introduction

Paragraph 1 is not needed, especially since mechanism of action is not well-described, it adds minimal value to the context. It is OK to open with Paragraph #2.

For Paragraph 2 in addition to indication listed, it is important to expand on the magnitude of the problem that AI-related adverse events lead to discontinuation of therapy.

Paragraph #3 - be specific which adverse effects are mitigated by exercise interventions and what type of exercise interventions. The current description seems very broad and again non-specific.

Paragraph 4 mixes data on muscle relaxation and exercise interventions. It is important to define the components on muscle relaxation exercises and the effect sizes seen in other populations to establish scientific plausibility that PRE would work in AI-induced arthralgias.

Methods

The details of the sample size calculation are inadequate – the authors should define what effect size they expected to see with a power of 80% and alpha of 0.05. A sample size of 20 per arm is a very small number for meaningful effect size.

Was the PRE implemented to the study participants as a group – in which case peer support will be a big confounder? I.e. did scores in pain and QALY improved due to the community/support aspect of the group and not necessarily the intervention in and of itself?

Should the analysis be intent to treat – patients excluded if they missed 10% of sessions? How did you establish fidelity of the at home sessions?

Not sure if statistical approach evaluating means is the appropriate approach, Eg. for table 3, authors assess mean in comparing scores on outcome instruments, but median and non-parametric analysis as they listed would have been most appropriate.

Results

Baseline characteristics should have also included stage of breast cancer as well as number of years on AI.

Table 3 – baseline scores for Table 3 for BPI-PS are higher for control group than for study group, and especially since the significance level is based on 1-point differences in report scores, I think this should be discussed as a limitation in interpreting effect size. Perhaps the model should have adjusted for baseline level of pain assessment. Results should also be interpreted in terms of what is a clinically significant difference between the groups – effect size between groups is not significant for any of the instruments. The instruments were described as assessing scores of 3-4 as mild, so is this study result clinically significant if the mean score @ 6 weeks for post groups fall within the mild range? In some cases, below the mild scale (<3).

Discussion:

The limitation section should be further expanded not a cursory statement that the sample size was small. Fidelity to home sessions should also be discussed. Authors should also discuss potential mediators of the effect noted. This should be an ITT analysis, is there a reason why the authors didn’t consider this?

Minor comments:

In the abstract, line 44 has an incomplete p value listed – 0.000.

6. PLOS authors have the option to publish the peer review history of their article (what does this mean?). If published, this will include your full peer review and any attached files.

Reviewer #1: No

Reviewer #2: **Yes: **Neven Patrick

Reviewer #3: No

---

## [Author Response · Author response to Decision Letter 0]

3 Jan 2024

We thank to editor and all reviewers for their valuable comments and contributions.

All requirements are outlined in the 'Response to Reviewers'.

---

## [Decision Letter · Decision Letter 1]

23 Feb 2024

PONE-D-23-36160R1Progressive Relaxation Training in Patients with Breast Cancer Receiving Aromatase Inhibitor Therapy-Randomized Controlled TrialPLOS ONE

Dear Dr. BAHÇACI,

Thank you for submitting your manuscript to PLOS ONE. After careful consideration, we feel that it has merit but does not fully meet PLOS ONE’s publication criteria as it currently stands. Therefore, we invite you to submit a revised version of the manuscript that addresses the points raised during the review process.

We look forward to receiving your revised manuscript.

Kind regards,

Peter Vuylsteke, MD

Academic Editor

PLOS ONE

Journal Requirements:

Additional Editor Comments:

Dear author

Thank you for the revisions made.

Kindly attend to the minor revision suggestion of Reviewer 1, then we can go on and accept the article.

Reviewers' comments:

Reviewer's Responses to Questions

**Comments to the Author**

1. If the authors have adequately addressed your comments raised in a previous round of review and you feel that this manuscript is now acceptable for publication, you may indicate that here to bypass the “Comments to the Author” section, enter your conflict of interest statement in the “Confidential to Editor” section, and submit your "Accept" recommendation.

Reviewer #1: (No Response)

2. Is the manuscript technically sound, and do the data support the conclusions?

Reviewer #1: Yes

3. Has the statistical analysis been performed appropriately and rigorously? 

Reviewer #1: Yes

4. Have the authors made all data underlying the findings in their manuscript fully available?

Reviewer #1: Yes

5. Is the manuscript presented in an intelligible fashion and written in standard English?

Reviewer #1: Yes

6. Review Comments to the Author

Reviewer #1: Minor revisions:

Table 2: In addition to frequencies, add the corresponding percentages.

7. PLOS authors have the option to publish the peer review history of their article (what does this mean?). If published, this will include your full peer review and any attached files.

Reviewer #1: No

---

## [Author Response · Author response to Decision Letter 1]

24 Feb 2024

Reviewer #1: Minor revisions:

Table 2: In addition to frequencies, add the corresponding percentages.

Authors’ Response: The authors kindly thank the reviewer for their feedback. The percentages of frequencies are now included in Table 2.

---

## [Editor Report · Decision Letter 2]

11 Mar 2024

Progressive Relaxation Training in Patients with Breast Cancer Receiving Aromatase Inhibitor Therapy-Randomized Controlled Trial

PONE-D-23-36160R2

Dear Dr. BAHÇACI,

We’re pleased to inform you that your manuscript has been judged scientifically suitable for publication and will be formally accepted for publication once it meets all outstanding technical requirements.

Kind regards,

Peter Vuylsteke, MD

Academic Editor

PLOS ONE

---

## [Editor Report · Acceptance letter]

8 Apr 2024

PONE-D-23-36160R2 

PLOS ONE

Dear Dr. Bahçacı, 

I'm pleased to inform you that your manuscript has been deemed suitable for publication in PLOS ONE. Congratulations! Your manuscript is now being handed over to our production team.

Kind regards, 

on behalf of

Dr. Peter Vuylsteke 

Academic Editor

PLOS ONE